# Collision Avoidance Path Planning for Automated Vehicles Using Prediction Information and Artificial Potential Field

**DOI:** 10.3390/s24227292

**Published:** 2024-11-14

**Authors:** Sumin Ahn, Taeyoung Oh, Jinwoo Yoo

**Affiliations:** 1Graduate School of Automotive Engineering, Kookmin University, Seoul 02707, Republic of Korea; asm2040@naver.com (S.A.); ohtaeyoung12@kookmin.ac.kr (T.O.); 2Department of Automobile and IT Convergence, Kookmin University, Seoul 02707, Republic of Korea

**Keywords:** autonomous driving, collision avoidance, path planning, artificial potential field, Bézier curve, sequential quadratic planning

## Abstract

With the advancement of autonomous driving systems, the need for effective emergency avoidance path planning has become increasingly important. To enhance safety, the predicted paths of surrounding vehicles anticipate risks and incorporate them into avoidance strategies, enabling more efficient and stable driving. Although the artificial potential field (APF) method is commonly employed for path planning due to its simplicity and effectiveness, it can suffer from the local minimum problem when using gradient descent, causing the vehicle to become stuck before reaching the target. To address this issue and improve the efficiency and stability of path planning, this study proposes integrating prediction data into the APF and optimizing the control points of the quintic Bézier curve using sequential quadratic planning. The validity of the proposed method was confirmed through simulation using IPG CarMaker 12.0.1 and MATLAB/Simulink 2022b.

## 1. Introduction

Autonomous driving technology has gained significant attention due to its potential to reduce traffic congestion, enhance driving safety, and provide greater convenience to drivers. As a result, numerous studies are actively exploring various aspects of this technology [1,2]. The rapid development of autonomous driving technology is driven by OEMs like Google Waymo, Tesla, and Uber, as well as academic researchers working in this field [3,4]. In response to these advancements, governments are preparing regulations and guidelines for the introduction of autonomous vehicles, while also enacting safety standards to ensure their safe deployment [5,6].

However, the rapid pace of autonomous driving also presents significant challenges. As self-driving cars enter commercialization, incidents involving collisions with human-driven vehicles have emerged, and these collisions continue to occur in notable numbers [7]. This has led to growing interest in autonomous driving systems capable of preventing potential collisions. Autonomous vehicles have demonstrated the ability to prevent collisions caused by human error in real-world traffic environments [8,9]. This capability is expected to significantly reduce accidents, improve road safety, and enhance traffic efficiency [10].

To prevent collisions, autonomous vehicles must be able to predict potential hazards and plan their movements accordingly. Collision prediction involves forecasting the future behavior of surrounding objects and assessing the risk of a collision with the autonomous vehicle [11]. Given the importance of utilizing predictive information about the behavior of nearby objects, numerous studies are focused on developing methods to accurately predict these behaviors [12,13,14,15]. While existing studies have primarily focused on improving the accuracy of behavior prediction, this research aims to overcome the limitations of traditional methods that rely on real-time assessment or short-term responses by utilizing predictive information directly in collision avoidance path planning. By proactively predicting potential collisions and, upon detection, integrating predictive data into an artificial potential field to identify safe positions, this approach enables the design of stable and efficient paths using Bézier curves. This method offers higher safety and reliability than real-time evaluation techniques such as Time-To-Collision (TTC). Also, this approach allows for effective collision avoidance even in complex road environments, contributing significantly to the driving safety of autonomous vehicles.

In summary, this paper proposes a path planning method that leverages prediction information about surrounding objects to avoid collisions. The proposed method integrates the IPG CarMaker simulation environment and the MATLAB/Simulink to perform real-time path planning, allowing the autonomous vehicle to avoid collisions efficiently. The approach combines artificial potential field (APF) with Bézier curve-based path generation and sequential quadratic programming (SQP) optimization to create an optimal, stable collision-avoidance path. Control inputs are seamlessly delivered from MATLAB/Simulink to the CarMaker simulation, enabling real-time execution.

## 2. Related Work

### 2.1. Traditional Path Planning Using Artificial Potential Field

The APF is a widely used path planning method originally developed in the field of mobile robotics. In APF-based path planning, a virtual potential field is created to guide a vehicle safely to its target point while avoiding obstacles [16]. The necessary data to construct this field come from sensors on autonomous vehicles, which perceive the surrounding environment. The potential field can be divided into three main categories: attractive potential field, repulsive potential field, and total potential field.

An attractive potential field generates a virtual force that pulls the vehicle toward the target point, while a repulsive potential field generates a force that pushes the vehicle away from obstacles or other vehicles. The repulsive potential field is calculated based on the position and speed of obstacles in the environment. The total potential field is the sum of the attractive and repulsive fields, guiding the vehicle along a path with the lowest potential value from its current location to the target point. However, the gradient descent method used for path planning in APF can lead to issues in complex driving environments. Specifically, the vehicle may encounter a local minimum where it gets stuck and cannot reach the target point, or experience oscillation in its trajectory. These limitations have prompted further research into improving APF-based path planning.

Several studies have addressed these issues [17,18,19,20,21]. Yanjun proposed a framework for autonomous vehicle path planning that utilizes a local electric comparison method to avoid collisions. In this approach, the driving area is divided into a mesh and repulsive values are assigned to each edge based on APF’s resistance approach, helping the vehicle find a collision-free path [22]. Lin introduced a waypoint tracking method that integrates collision avoidance using APF theory. In this method, the APF-generated path is optimized before and after collision avoidance to ensure a smooth connection with waypoints obtained via GPS, preventing sudden changes in the vehicle’s heading angle during collision maneuvers [23].

Additionally, research is being proposed that focuses not just on collision avoidance but also on mitigating the impact of collisions where avoidance is impossible. To address this, Xu Shang developed a safe controller using model predictive control, which enhances the APF method by incorporating the shape information of surrounding roads and vehicles. This approach provides a more practical solution for emergency scenarios and offers a methodology for reducing the severity of collisions, particularly in specific parts of the vehicle [24].

### 2.2. Bézier Curve Trajectory Generation

The Bézier Curve is a widely used technique in path planning, particularly in autonomous driving and robotics. Its ability to generate smooth and safe paths has made it an integral part of several path planning algorithms [25,26,27]. The Bézier Curve is defined by control points, with the shape of the curve determined by their positions. Mathematically, the Bézier Curve is based on the Bernstein polynomials, and the Nth Bézier Curve is defined by n+1 control points. The equation for the Nth Bézier Curve is given by:(1)$$\begin{array}{c} B\left(t\right)=\left(1-t\right)\sum_{i=0}^{n}b_{i,n}\left(t\right)P_{i}+t\sum_{i=0}^{n}b_{i,n}\left(t\right)P_{i}\\ =\sum_{i=0}^{n}\frac{n+1-i}{n+1}b_{i,n+1}\left(t\right)P_{i}+\sum_{i=0}^{n}\frac{i+1}{n+1}b_{i+1,n+1}\left(t\right)P_{i}\\ =\sum_{i=0}^{n+1}\left(\frac{i}{n+1}P_{i-1}+\frac{n+1-i}{n+1}P_{i}\right)b_{i,n+1}\left(t\right)\\ =\sum_{i=0}^{n+1}b_{i,n+1}\left(t\right)P_{i}^{,}\\ \left(P_{i}^{,}=\frac{i}{n+1}P_{i-1}+\frac{n+1-i}{n+1}P_{i},i=0,\ldots ,n+1\right) \end{array}$$

The starting and ending points of the curve are fixed by the first and last control points, while the shape of the curve changes based on the position of the control points. Bézier curves have the advantage of providing sufficient degrees of freedom while ensuring the creation of smooth and continuous paths. Paths based on cubic, quartic, and quintic Bézier curves are shown in Figure 1.

Given their advantages, Bézier curves have been employed in various studies for autonomous vehicle path planning. For instance, J. Chen proposed a continuous curvature path using a partial second-order Bézier curve that meets the requirements for smoothness. This was achieved by applying minimum lane change distance calculations and curvature constraints to the path planning [28].

J. Moreau, using a nonlinear dynamics vehicle model, constrained the Bézier curve to avoid obstacles in real time. In the case of unexpected events or obstacles, such as those encountered at intersections, the path is optimized using the quadratic programming (QP) method to generate the shortest avoidance path. This method produced smoother paths than those generated by gradient descent in APF while ensuring stable steering angles and faster computation times [29].

C. Chen proposed an algorithm to ensure continuous curvature and bounded curvature trajectories for autonomous vehicles from the initial to the target state. To achieve this, the quartic Bézier curve was adopted, with the Bézier curve’s parameters being continuous and bounded using the SQP method [30]. This algorithm not only ensures a continuous and smooth path but also provides a collision avoidance path planning approach through the optimization of control points.

### 2.3. Autonomous Driving System Based on Vehicle Trajectory Prediction

An autonomous driving system that utilizes predicted trajectories can significantly enhance safety by detecting potential risks in advance and formulating appropriate driving strategies to mitigate them. The road environment encountered by autonomous vehicles is highly dynamic, making it challenging to respond to the movements and actions of other vehicles in real time. Unexpected situations, such as sudden lane changes, speed variations, or the sudden appearance of obstacles from surrounding vehicles, are common. To respond effectively in such situations, autonomous vehicles must predict the future paths of surrounding vehicles and plan their path accordingly, based on this predicted information [31,32,33,34].

A notable example of this approach is the work of D. Meng, who proposes a method for autonomous vehicles to predict the trajectories of surrounding vehicles in real time to identify potential risks. This approach gathers data on the current state of nearby vehicles, including location, speed, and acceleration, using various sensors and algorithms. It then predicts the future movement of these vehicles based on the collected data. Using the predicted path, the autonomous vehicle calculates the potential risk of collision using metrics like TTC and minimum distance margin. A methodology has been developed where the vehicle can decide whether to maintain its current lane or change lanes to avoid risk based on the calculated collision probability [35]. J. Kim further argued that previous algorithms relying on TTC often assumed constant conditions and inputs. To address this, an improved risk prediction algorithm is proposed that offers more accurate TTC predictions by forecasting the future trajectories of surrounding vehicles. This algorithm also proposes an inverse TTC map to enhance the prediction accuracy [36].

The primary advantage of autonomous driving systems based on predicted trajectories is enhanced safety. These systems are far more efficient than traditional reaction-based systems because they can recognize risk factors before unexpected events occur. This proactive approach allows the system to respond preemptively. For example, if an autonomous driving system predicts that the vehicle in front will suddenly change lanes, it can adjust speed or change lanes in advance, planning a safer path. Such preemptive responses can significantly reduce accidents caused by sudden braking or other reactive maneuvers in unexpected situations, helping to maintain a safe distance between vehicles.

## 3. Proposed Collision-Avoidance Path Planning Method

### 3.1. Overall Architecture of the Proposed Method

This study focuses on a path planning methodology that emphasizes the use of prediction information to avoid collisions with nearby vehicles. The study operates under the assumption that accurate prediction data is received, which is provided through pre-generated simulation data spanning approximately 2 s into the future with a time interval of 0.1 s. Although it is difficult to receive accurate prediction data in real-world without considering uncertainties, we believe that receiving the prediction data at the minimum time lengths makes it sufficiently understandable for the purpose of developing the path planning algorithm. This 2 s prediction horizon is commonly accepted as the minimum necessary time frame in trajectory prediction research, making it an ideal choice for our study. Architecture of the proposed methodology determines whether the avoidance path generation algorithm operates by conducting a collision assessment based on the predicted path of the surrounding vehicle and the planned path of the ego vehicle, as shown in Figure 2. When a collision with a surrounding vehicle is predicted, the APF identifies the point with the lowest potential value and designates it as the target point. A path for collision avoidance is then created using a quintic Bézier curve, optimized based on a cost function that considers factors such as curvature, total potential value, jerk, and the lateral offset from the current position to the target point.

### 3.2. Risk Assessment

When driving along the planned path of an autonomous vehicle, risk can be measured using a simplified kinematic formula that considers the time domain, distance, and acceleration domains. Key indicators such as TTC and time headway in the time domain, minimum safe distance in the distance domain, and steering threat number in the acceleration domain are used to assess risk [37,38,39,40]. However, these risk assessment methods are based on the current state of the vehicle, making it difficult to predict future states. As a result, these indicators may not provide sufficient response time during emergency avoidance maneuvers or urgent situations. If the future behavior of surrounding objects can be predicted, autonomous vehicles will have more time to react to potential risks in critical situations. To address this, this study employs risk assessment based on collision detection using circular boundaries, enabling faster response times by utilizing future prediction information of surrounding objects. 

The collision assessment circle is designed to cover the body of the vehicle. For the ego vehicle, its specifications are known. However, for surrounding traffic, the sensor of the ego vehicle can only measure the overall width and length. Therefore, the collision assessment circle for a surrounding vehicle is defined accordingly, as shown in Figure 3. The points $x_{Ego}$ and $y_{Ego}$ represent the center points of the ego vehicle’s rear axle. For the ego vehicle, two collision assessment circles are defined, and their center positions and radii are calculated as follows:(2)$$R_{Ego}=\frac{1}{2}\sqrt{{\left(\frac{L_{R}+L_{w}+L_{F}}{2}\right)}^{2}+{L_{B}}^{2}},$$
(3)$$x_{Ego\_f}\left(t\right)=x_{Ego}\left(t\right)+\frac{1}{4}\left(3L_{w}+3L_{F}-L_{R}\right)cos\theta_{Ego}\left(t\right),\;y_{Ego\_f}\left(t\right)=y_{Ego}\left(t\right)+\frac{1}{4}\left(3L_{w}+3L_{F}-L_{R}\right)sin\theta_{Ego}\left(t\right),\;x_{Ego\_r}\left(t\right)=x_{Ego}\left(t\right)+\frac{1}{4}\left(L_{w}+L_{F}-3L_{R}\right)cos\theta_{Ego}\left(t\right),\;y_{Ego\_r}(t)=y_{Ego}\left(t\right)+\frac{1}{4}\left(L_{w}+L_{F}-3L_{R}\right)sin\theta_{Ego}(t).$$

For surrounding traffic, three collision assessment circles are defined, with their center positions and radii determined as follows:(4)$$R_{trf}=\frac{W_{trf}}{2sin\left(\frac{\pi }{4}\right)},$$
(5)$$x_{trf\_f}\left(t\right)=x_{trf}\left(t\right)+L_{trf}cos\theta \left(t\right)-R_{trf}\cos\left(\frac{\pi }{4}\right)cos\theta_{trf}\left(t\right),\;y_{trf\_f}\left(t\right)=y_{trf}\left(t\right)+L_{trf}sin\theta \left(t\right)-R_{trf}\cos\left(\frac{\pi }{4}\right)sin\theta_{trf}\left(t\right),\;x_{trf\_r}\left(t\right)=x_{trf}\left(t\right)+R_{trf}\cos\left(\frac{\pi }{4}\right)cos\theta_{trf}\left(t\right),\;y_{trf\_r}\left(t\right)=y_{trf}\left(t\right)+R_{trf}cos\left(\frac{\pi }{4}\right)sin\theta_{trf}\left(t\right),\;x_{trf\_c}\left(t\right)=\frac{x_{trf\_f}\left(t\right)+x_{trf\_r}\left(t\right)}{2},\;y_{trf\_c}\left(t\right)=\frac{y_{trf\_f}\left(t\right)+y_{trf\_r}\left(t\right)}{2}.$$

To check for collision between the defined circles of the ego vehicle and those of surrounding objects, the following method is used. In this study, collisions are assessed at each time step using the ego vehicle’s planned path and the predicted trajectories of surrounding vehicles for the next two seconds. The collision assessment is performed by comparing the center points and radii of the defined circles based on the following formula.
(6)$$\left\|C_{j}^{i}-C_{l}^{i}\right\|=\sqrt{{\left(x_{j}^{i}-x_{l}^{i}\right)}^{2}+{\left(y_{j}^{i}-y_{l}^{i}\right)}^{2}},i=1,2,\ldots ,n\;\mathrm{Risk}\;=\;1,\;\;\;\;\mathrm{if}\;\left\|C_{j}^{i}-C_{l}^{i}\right\|\leq R_{\mathrm{ego}}+R_{\mathrm{trf}}$$
where $C_{j}^{i}$ represents the center point of the circle at time step i on the ego vehicle’s planned path, j denotes either the front or rear of the vehicle, $C_{l}^{i}$ represents the center point of the circle at the time step i on the predicted path of the surrounding vehicle, and l denotes the front, center, or rear of the vehicle. If the distance between the center points is smaller than the sum of the radii of the two circles, a collision is predicted. The parameters used to define the risk circle are provided in Table 1.

### 3.3. Artificial Potential Field with Prediction Information

The APF is a path-planning algorithm that generates a virtual attractive potential field and a repulsive potential field to guide a vehicle to its target. In this study, we consider the surrounding environment by constructing a suitable potential function that creates a road potential field (considering the lanes boundaries) and an obstacle potential field (considering static or moving obstacles). Traditional APF designs an attractive potential field to lead the vehicle toward the goal, but in the proposed method, we eliminate the attractive potential field. Instead, we designate the point with the lowest potential value as the target point in APF. The road potential field is used to keep the vehicle in the center of the lane, preventing it from veering off-road, while the obstacle potential field ensures the vehicles avoid collisions with obstacles. The combined values of these two potential fields are represented as follows:(7)$$U_{total}=U_{Road}+U_{Obstacle}.$$

The road potential field is applied to represent a three-lane road, as illustrated in Figure 4a, using trigonometric functions. The potential field function is defined in Equation (8), and Figure 4b shows a 3D representation of the road potential field based on the following equation:(8)$$A\left(Y\right)=\frac{\left|sgn\left(Y-Y_{c1}\right)+sgn\left(Y-Y_{c3}\right)\right|\left(1-p\right)+2p}{2},\;U_{Road}=\frac{A\left(Y\right)\{\cos\left(\frac{2\pi Y}{w}\right)+1\}}{2}.$$
where w is the width of the road, $Y_{c1}$ and $Y_{c3}$ are the center positions of the first and third lanes, respectively, in the road’s transverse coordinate system, and p is the coefficient used to generate values between 0 and 1.

The obstacle potential field incorporates both static and moving obstacles and is designed to help autonomous vehicles maintain a safe distance from them. As the autonomous vehicle approaches an obstacle, the potential value increases gradually, preventing a collision. The obstacle potential field is expressed as follows:(9)$$U_{obstacle}^{i}=\left\{\begin{array}{l} e^{-\left[\left\{\frac{1}{\left(\left|v_{ego}-v_{obs}^{i}\right|+\mu \right)}\right\}\left\{\frac{{\left(x-x_{obs}^{i}\right)}^{2}}{\delta_{x}^{2}}-\frac{{\left(y-y_{obs}^{i}\right)}^{2}}{\delta_{y}^{2}}\right\}\right],\;\;\;if\;\left(v_{ego}\geq v_{obs}^{i}\;\&\;x\leq x_{obs}^{i}\right)\;or\;\left(v_{ego}<v_{obs}^{i}\;\&\;x>x_{obs}^{i}\right)}\\ 0,\;\;\;\;\;else \end{array}\right.,$$
where μ is a small constant added to prevent computational errors when the speed of a surrounding object is the same as that of the ego vehicle; $\delta_{x}$ and $\delta_{y}$ are standard deviations in the longitudinal (x) and lateral (y) directions, respectively, affecting the size of the potential field. Figure 5 shows how the obstacle potential field changes based on different values of $\delta_{x}$ and $\delta_{y}$.

Figure 6 presents both 3D and top-down views of the combined road potential and obstacle potential fields. The ego vehicle is positioned at [0, −5.25] and is traveling at 80 km/h. The vehicle ahead is located at [40, −5.25] and is moving at 60 km/h, while the vehicle in the right-front position is located at [10, −8.75] and moving at 70 km/h. The front vehicle poses a greater threat, as indicated by its larger area, due to its higher relative velocity compared to the right-side vehicle. The parameters used for the potential fields are outlined in Table 2. These parameters used in this study were determined through a trial-and-error process to achieve optimal suitability within the experimental environment. In real-world scenarios, however, these parameters may not remain fixed.

In this study, when planning a collision-avoidance path, the point with the lowest potential value in the APF is identified and designated as the target point. If the future position of surrounding objects is not considered when determining the target point, additional collisions may occur during the avoidance maneuver. To address this, the future position and speed of surrounding vehicles are incorporated into the APF using prediction information of other surrounding objects. When considering prediction data for the next 2 s, it is impractical to use all time steps, as the intended target point may not be found, and integrating all information could also lead to increased computational burden, potentially reducing overall system performance. Therefore, the 2 s prediction data is divided into n time steps and incorporated into the APF. When reflecting the future obstacle potential field in the APF, the values of $\delta_{x}$ and $\delta_{y}$ are adjusted accordingly.

In this study, the future prediction is divided into five time steps, including the present and data from the next 2 s. Figure 7 shows an example of the top-down view and 3D visualization of the total potential field, combining both the obstacle and road potential fields, with the future position of the front vehicle located at [40, −5.25].

### 3.4. Path Generation and Optimisation Based on the Bézier Curve

The proposed collision-avoidance path planning algorithm is based on the Bézier curve. In this approach, the positions of the current start point and target point are fixed, while the path is generated by adjusting the positions of the remaining control points. The Bézier curve described in Section 2.2 follows Equation (1), where $P_{i}$ represents the position of the control points. The start and end points of the control point are defined as follows:(10)$$P_{0}=\left[x_{start}\;,\;y_{start}\right],\;P_{n}=[x_{target}\;,\;y_{target}].$$

Figure 8 illustrates the selection of a target point. The black line is selected based on the target longitudinal distance. In this study, $x_{target}=v_{ego}\times t$, where t is set to 1.5 s. Once $x_{target}$ is determined, the potential value at that target state in the APF is checked, and the point with the lowest potential value is selected as $y_{target}$.

In this algorithm, the path is planned based on the vehicle’s local coordinate system. The conversion from the global coordinate system to the vehicle’s local coordinate system is given as follows:(11)$$\left[\begin{array}{c} x_{local}\\ y_{local} \end{array}\right]=\left[\begin{array}{cc} cos\theta & -sin\theta \\ sin\theta & cos\theta \end{array}\right]\left[\begin{array}{c} x_{global}-x_{ego}\\ y_{global}-y_{ego} \end{array}\right].$$

#### 3.4.1. Quartic Bézier Curve Modeling

In the case of the quartic Bézier curve, there are three adjustable control points, and the resulting curve is shown in Figure 9a. The first and last control points follow Equation (10), while the remaining three control points are defined as follows:
(12)$$P_{0}={\left[0,0\right]}^{T},\;P_{1}={\left[l_{1},0\right]}^{T},\;P_{2}={\left[x_{2},\frac{4\kappa_{0}{l_{1}}^{2}}{3}\right]}^{T},\;{\;\kappa }_{0}=\frac{3y_{2}}{4l_{1}^{2}},\;{P_{3}=[x_{4}-l_{2}cos\varphi_{T},\;y_{4}-l_{2}sin\varphi_{T}]}^{T},\;P_{4}={\left[x_{4},y_{4}\right]}^{T}.$$
where $P_{1}$ is located at a distance of $l_{1}$ along the longitudinal direction from $P_{0}$, $P_{2}$ is positioned considering the initial curvature value $\kappa_{0}$, and $P_{3}$ is positioned using the distances $l_{2}$ and the angle $\varphi_{T}$. The distance between $P_{3}$ and $P_{4}$ is modeled accordingly.

#### 3.4.2. Quintic Bézier Curve Modeling

In the case of quintic Bézier curve, four control points are adjustable, and the modeling result is shown in Figure 9b. The first and last control points follow the same rule as in the quartic Bézier curve. The remaining four control points are defined as follows:(13)$$P_{0}={\left[0,0\right]}^{T},\;P_{1}={\left[l_{1},0\right]}^{T},\;P_{2}={\left[l_{1}+l_{2},0\right]}^{T},\;{P_{3}=[x_{4}-l_{3}cos\varphi_{T},y_{4}-l_{2}sin\varphi_{T}]}^{T},\;P_{4}={\left[x_{5}-l_{4}cos\varphi_{T},y_{5}-l_{4}sin\varphi_{T}\right]}^{T},\;P_{5}={\left[x_{5},y_{5}\right]}^{T}$$
where $P_{1}$ is located $l_{1}$ units in the longitudinal direction from $P_{0}$, $P_{2}$ is located $l_{1}+l_{2}$ units in the longitudinal direction, and $P_{3}$ and $P_{4}$ are calculated based on the distances $l_{3}$, $l_{4}$ and the angle $\varphi_{T}$ relative to previous control points.

#### 3.4.3. Path Optimization

To ensure that the designed Bézier curve model is collision-free, smooth, and continuous, an optimization technique is required to find valid parameters for each model. SQP converts nonlinear problems—incorporating objective functions and constraints—into QP problems by linearizing or approximating them. The QP problem is solved iteratively to obtain an approximate solution, with each iteration refining the approximation of the objective function while ensuring that the constraints are satisfied. SQP converges quickly since it utilizes the Hessian matrix, based on Newton’s method, and efficiently handles various constraints during path optimization [41,42,43,44].

In this study, the objective function is designed to balance driving convenience and safety while ensuring collision-free operation of autonomous vehicles. Path continuity and smoothness are critical factors in enhancing the comfort of vehicle occupants. A path that is not smooth or continuous can lead to discomfort and may affect the stability of the vehicle’s behavior. Moreover, the vehicle must avoid collisions with surrounding objects. The total potential field, as discussed earlier, ensures the safety of the planned path while also minimizing the lateral distance to the target point. This design considers situations where the autonomous vehicle must make rapid evasive maneuvers while following the planned path. The objective function J that considers these features is expressed as follows:(14)$$J=a\int_{0}^{1}\kappa^{2}\left(\tau \right)d\tau +b\int_{0}^{1}U_{total}\left(\tau \right)d\tau +c\int_{0}^{1}Jerk\left(\tau \right)d\tau +d\int_{0}^{1}D_{y}\left(\tau \right)d\tau$$
$$\mathrm{subject}\;\mathrm{to}\;:\left|\kappa {\left(\tau \right)}_{max}\right|\leq \kappa_{max}\;\left\{\begin{array}{l} \left[l_{1},x_{2},l_{2}\right]>0,\;if\;Bé\mathrm{zier}\;\mathrm{order}=4\\ \left[l_{1},l_{2},l_{3},l_{4}\right]>0,\;if\;Bé\mathrm{zier}\;\mathrm{order}=5\; \end{array}\right.$$
where κ represents the curvature, and $U_{total}$, Jerk, and $D_{y}$ represent the potential value of the planned path, jerk, and lateral offset from the target point, respectively. The objective function J, structured as described, optimizes the position of each control point through parameters such as $l_{1},l_{2}$ and others.

The term that minimizes the lateral distance from the point-of-view function to the target point enables rapid avoidance behavior during path optimization. To account for this, the TTC with potential obstacles is considered. TTC is a measure of the time remaining before a collision occurs between the ego vehicle and a potential obstacle. It is defined as the ratio of the relative distance to the target obstacle to the relative velocity towards the target obstacle. This metric helps to assess the urgency of collision avoidance and provides a basis for triggering avoidance actions. The definition of d, which represents coefficient of laterally offset term, is given by Equations (15) and (16):(15)$$\mathrm{TTC}=\frac{RD_{target}}{RV_{target}},$$
(16)$$d=\left\{\begin{array}{l} \varepsilon ,\;\;\;if\;\mathrm{TTC}<\gamma \\ 0,\;\;\;else \end{array}\right..$$

$RD_{target}$ is the relative distance to target vehicle and $RV_{target}$ is the relative velocity towards the target vehicle. The parameters used to construct the objective function are listed in Table 3.

## 4. Simulation Results

The proposed methodology aims to use prediction information from surrounding traffic to efficiently and stably avoid collision. We evaluate this methodology by comparing scenario results with and without the prediction information, as well as using quartic and quintic Bézier curves. Four test scenarios are used to evaluate the methodology. Scenarios A to C are based on autonomous systems test scenarios defined by ISO DIS 34502 [45], as shown in Figure 10, and Scenario D is a test scenario for continuous avoidance behavior. Scenarios A to C run for 13 s, while Scenario D runs for 15 s. The specifications of each scenario will be described at the beginning of each experiment’s results section. The collision avoidance path algorithm is triggered based on a TTC threshold of less than 2 s when prediction information is not used in order to allow comparison with the system that utilizes prediction information.

To validate the proposed avoidance path generation methodology, an integrated environment, as shown in Figure 11, is set up using IPG’s software, CarMaker. The driving road information, surrounding vehicle data, and perception results used in the avoidance path generation algorithm are provided by CarMaker. When a collision is predicted during the ego vehicle’s driving, the proposed algorithm plans the optimal avoidance path. This planned path is then passed to the pure pursuit lateral controller, where the steering angle is calculated and subsequently transmitted to the CarMaker vehicle model. This integrated environment is configured using IPG CarMaker version 12.0.1 and MATLAB 2022b/Simulink, ensuring smooth integration and compatibility with other software.

This platform allows real-time monitoring of the vehicle’s behavior during testing and simulates realistic vehicle dynamics through parameterized components, such as steering, tires, brakes, powertrain, and chassis. The vehicles used in this study are those provided by IPG CarMaker. The ego vehicle is a Hyundai-IONIQ 5, and the traffic vehicle is an IPG_CompanyCar_2018. Simulation results support Simulink integration, and the driving results from each collision avoidance path planning system are evaluated using three methods.

Driving trajectory—to visualize the overall driving situation for each scenario.Steering wheel angle, yaw rate, and lateral acceleration plots—to assess the vehicle’s lateral stability.Maximum value analysis—to examine the system’s lateral stability during the avoidance maneuver.

### 4.1. Simulation Scenario A

In Scenario A, as shown in Figure 10a, the ego vehicle is driving in the second lane of a third-lane road at 80 km/h, while the traffic vehicle, 40 m in front in the same lane, is also driving at 80 km/h. At this point, if the traffic vehicle rapidly accelerates and decelerates at −6 m/s^2^, the ego vehicle must perform a collision avoidance maneuver by changing lanes safely and smoothly.

The simulation results are shown in Figure 12 and Figure 13, illustrating the driving situations at 0, 3, 6, 6.5, 9, 11, and 13 s, depending on whether the prediction is utilized and whether the quartic or quintic Bézier curve algorithm is applied. The results include changes in the ego vehicle’s steering wheel angle, yaw rate, and lateral acceleration over time. Figure 12 shows the driving trajectories at each time interval, confirming that the system utilizing prediction started avoidance earlier. At 6 s, the driving paths of the system with prediction, represented in red and blue, secured more lateral distance from the traffic vehicle compared to the paths of the system without prediction, shown in yellow and purple. Figure 12b illustrates the driving situation at 6.5 s in Scenario A, where the red trajectory represents the use of prediction with the quartic Bézier algorithm, while the yellow trajectory shows the result without prediction. At this point, using prediction allowed the system to begin avoidance earlier, achieving 0.363 m more lateral distance compared to the system without prediction. The quartic Bézier algorithm, without prediction, measured a lateral distance of 3.01 m from the traffic. Similarly, the blue path in Figure 12c represents the quintic Bézier algorithm with prediction, which achieved 0.235 m more lateral distance compared to the algorithm without prediction, which measured 2.96 m from the traffic.

The advantage of using prediction is that it enables early recognition of the surrounding environment, allowing for quicker preparation for potential dangers. This ultimately enhances vehicle stability. Figure 13 displays graphs representing the steering wheel angle, yaw rate, and lateral acceleration for each system in Scenario A. Upon examining the graphs, it is evident that the system utilizing prediction planned a smoother path, resulting in improved lateral stability. Additionally, the quintic Bézier curve provides more stable lateral behavior than the quartic Bézier curve. As shown in Table 4, when comparing the maximum values, the system using prediction recorded smaller maximum values than the system without prediction. Furthermore, the quintic Bézier algorithm achieved smaller maximum values compared to the quartic Bézier algorithm.

### 4.2. Simulation Scenario B

Scenario B, illustrated in Figure 10b, involves the ego vehicle driving at 80 km/h in the second lane of a three-lane road. A traffic vehicle in the left lane, located 15 m ahead, is driving at 75.6 km/h and begins a cut-in maneuver into the ego vehicle’s lane with a deceleration rate of −1.5 m/s^2^. This type of cut-in scenario is common in real-world driving and often occurs unexpectedly, making it an ideal situation to evaluate collision avoidance maneuvers. In this scenario, the ego vehicle must not only avoid a collision but also smoothly execute a lane change to the right while avoiding the traffic vehicle on the left.

The simulation results are shown in Figure 14 and Figure 15, depicting the driving situation of the vehicles at 0, 3, 6, 6.5, 9, 11, and 13 s, along with changes in the ego vehicle’s steering wheel angle, yaw rate, and lateral acceleration over time. Figure 14 shows the vehicle paths at each time interval, confirming that the system using prediction initiates avoidance maneuvers earlier. At 4.3 s, the driving paths for the system with prediction, represented by the red and blue trajectories, show a significantly greater lateral distance from the traffic compared to a system without prediction, depicted by yellow and purple trajectories. Figure 14b,c illustrate the driving situation at 5.2 s in Scenario A. The red path represents the use of prediction with the quartic Bézier algorithm, while the yellow path shows the outcome without prediction. With prediction, the system initiates the evasive maneuver 0.6 s earlier, achieving a greater lateral distance from the traffic vehicle. The red path, using prediction, maintained 1.26 m more lateral distance compared to the yellow path, which did not use prediction. The quartic Bézier algorithm without prediction resulted in a lateral distance of 2.42 m from the traffic vehicle. Similarly, the blue path in Figure 14c represents the quintic Bézier algorithm with prediction, which secured 1.53 m more lateral distance than the algorithm without prediction. The algorithm without prediction resulted in a lateral distance of 2.19 m from the traffic vehicle.

Figure 15 presents the graphs illustrating the steering wheel angle, yaw rate, and lateral acceleration for each system in Scenario B. From the graphs, it is clear that the system incorporating prediction generated a smoother path and achieved greater lateral stability. Additionally, the quintic Bézier curve demonstrated more stable lateral performance than the quartic Bézier curve. As detailed in Table 5, since Scenario B involves a more abrupt maneuver than Scenario A, the maximum values for lateral acceleration, yaw rate, and steering wheel angle are higher. However, the system with prediction consistently recorded smaller maximum values than the system without prediction. Furthermore, comparing the two algorithms, the quintic Bézier algorithm consistently resulted in smaller maximum values than the quartic Bézier algorithm. For instance, the highest recorded lateral acceleration in Scenario B was 0.5574 g, the yaw rate was 0.2429 deg/s, and the steering wheel angle pealed at 44.6627 deg, all of which occurred when using the quartic Bézier algorithm without prediction.

### 4.3. Simulation Scenario C

Scenario C, illustrated in Figure 10c, involves the ego vehicle driving in the second lane of a three-lane road at 80 km/h. Ahead in the same lane, there are two vehicles: Traffic1, positioned 20 m ahead, driving at 82.8 km/h, and Traffic2, 40 m ahead, driving at 72 km/h. At this point, Traffic1 initiates a cut-out maneuver into the right lane with a deceleration rate of −1.5 m/s^2^. While Traffic1 is cutting out, the Traffic2 vehicle decelerates at a rate of −3 m/s^2^. This scenario simulates a situation where, as the lead vehicle (Traffic1) moves to another lane, a new vehicle (Traffic2) appears and begins decelerating. In this case, the ego vehicle must quickly reassess the updated surroundings, select an appropriate lane to avoid the potential hazard, and execute a safe maneuver.

The simulation results are presented in Figure 16 and Figure 17, showing the positions of the vehicles at 0, 2.5, 5.2, 5.6, 7.5, 9, 11, and 13 s, along with the changes in the ego vehicle’s steering wheel angle, yaw rate, and lateral acceleration over time. Figure 16 illustrates the vehicle trajectories at each time point, confirming that the system using prediction initiated the avoidance maneuver first. At 5.2 s, Traffic2 can be seen stopped in its original lane, and the trajectories of the system with prediction (red and blue) show a much greater lateral distance from the traffic compared to the system without prediction (yellow and purple). Figure 16b,c display the evasive driving trajectories at 5.6 s in Scenario C for each algorithm. The red and yellow trajectories represent the quartic Bézier algorithm with and without prediction, respectively. Using prediction, the system was able to predict the collision and initiate the avoidance maneuver 0.5 s earlier than the system without prediction, achieving a greater lateral distance from Traffic2. The red trajectory with prediction secured 0.73 m more lateral distance compared to the yellow trajectory without prediction, which showed a lateral distance of 2.59 m from Traffic2. Similarly, the blue trajectory in Figure 16c represents the quintic Bézier algorithm with prediction, which secured 0.7 m more lateral distance than the system without prediction, which had a lateral distance of 2.54 m from Traffic2.

Figure 17 displays the graphs of the steering wheel angle, yaw rate, and lateral acceleration for each system in Scenario C. The results indicate that the system utilizing prediction had more time for collision avoidance, allowing for a smoother path and greater lateral stability compared to the system without prediction. Additionally, the quintic Bézier curve ensures a more stable lateral motion than the quartic Bézier curve. As shown in Table 6, since Scenario C involves smoother maneuvers than Scenario A, lower maximum values were recorded. Consistent with previous scenarios, the system using prediction recorded smaller maximum values than the system without prediction. Moreover, the quintic Bézier algorithm resulted in smaller maximum values than the quartic Bézier algorithm, both with and without prediction.

### 4.4. Simulation Scenario D

Scenario D, illustrated in Figure 10d, involves the ego vehicle driving at 80 km/h in the second lane of a three-lane road. Ahead of the ego vehicle, Traffic2 is driving at 72 km/h, positioned 40 m ahead. In the left lane, Traffic1 is driving at 80 km/h, and in the right lane, Traffic3 is also driving at 80 km/h. Throughout this scenario, Traffic3 maintains constant velocity. Traffic2 begins to decelerate at a rate of −3 m/s^2^, forcing the ego vehicle to initiate an evasive maneuver by changing to the left lane, as it offers the most stable position relative to the surrounding traffic. The ego vehicle must safely and efficiently execute this lane change to avoid a potential collision. Following the successful completion of the first evasive maneuver, Traffic1 suddenly decelerates at a sharper rate of –6 m/s^2^, prompting the ego vehicle to quickly plan another lane change to the right lane. The ego vehicle must execute this second evasive maneuver promptly and safely, ultimately returning to the second lane. This scenario is designed to test the vehicle’s ability to perform continuous collision avoidance maneuvers and evaluate whether the system can carry out these successive maneuvers both stably and efficiently. It aims to validate the system’s ability to manage consecutive evasive maneuvers while maintaining safety and operational effectiveness.

The simulation results are presented in Figure 18 and Figure 19, showing the driving situations of the vehicles at 0, 3, 5, 5.85, 11, 13, and 15 s, along with the changes in the ego vehicle’s steering wheel angle, yaw rate, and lateral acceleration over time. In Figure 18, the vehicle positions at each time step are displayed, and it is evident that for both collision avoidance maneuvers, the system using prediction initiates the evasive actions earlier than the system without prediction. At 5 and 11 s, the system with prediction anticipates the potential collision sooner, resulting in earlier avoidance maneuvers. The red and blue trajectories (representing the predicted system) maintain a greater lateral distance from Traffic2 compared to the yellow and purple trajectories (representing the non-predicted system). This demonstrates the benefit of prediction in enabling earlier and more effective collision avoidance.

Figure 18b,c illustrate the evasive maneuvers at 5.85 and 11.25 s, respectively, in Scenario D. The system with prediction starts both evasive maneuvers 0.7 s earlier than the system without prediction, leading to a greater lateral distance from Traffic2. At 5.85 s, the red (predicted) trajectory achieves 0.92 m more lateral distance than the yellow (non-predicted) trajectory, which measures a distance of 2.47 m from Traffic2. In Figure 18b, the blue (predicted) trajectory secures 1 m more lateral distance than the purple (non-predicted) trajectory, which measures 2.25 m from Traffic2. Following the first evasive maneuver, Figure 18c shows that the system using prediction also achieves a greater lateral distance from Traffic1 during the second evasive maneuver, further demonstrating the advantage of prediction in collision avoidance.

Figure 19 shows graphs depicting steering wheel angle, yaw rate, and lateral acceleration for each system in Scenario D. Upon reviewing the graphs, it becomes evident that in both evasive maneuvers, the system utilizing prediction initiates the evasive actions earlier, allowing for more time to perform the maneuvers. This leads to more efficient path planning and improved lateral stability compared to the system without prediction. Table 7 demonstrates that the maximum values recorded in Scenario D were the highest among all previous scenarios. This is likely because, after the first collision avoidance maneuver, the vehicle did not fully stabilize before executing the subsequent evasive maneuvers, leading to larger recorded values. The results still show that, as with previous scenarios, the system using prediction achieved smaller maximum values than the system without prediction. Furthermore, when comparing both algorithms, the quintic Bézier algorithm consistently demonstrated smaller maximum values, indicating superior stability in generating safer paths.

## 5. Conclusions

This study proposes a collision avoidance path planning method that incorporates the predicted information from surrounding vehicles into the generation of avoidance paths. The methodology is grounded in the potential field approach used in the APF, where the ego vehicle selects the target point deemed safest for collision avoidance maneuvers. This process is divided into three stages to achieve effective collision avoidance.

In the first stage, the predicted information of surrounding vehicles, along with the ego vehicle’s planned path, is used to assess potential collisions by checking for intersections between the predefined risk circles of the ego and surrounding vehicles at each timestep. This predicted information includes the position, yaw, and velocity for the next 2 s from the current time. In the second stage, based on APF theory, the road potential field is constructed using environmental information, while the obstacle potential field is generated by considering the relative distance and velocity between the ego vehicle and other objects. To avoid additional collisions during the avoidance maneuver, future obstacle information is also factored in to create a future obstacle potential field. The road and obstacle potential fields are then combined to form the total potential field. The ego vehicle selects the point with the lowest potential value within the total potential field as the target point. In the final stage, the designated target point, along with the ego vehicle’s current position, is used as a control point to form an objective function that considers factors such as curvature, jerk, the sum of potential values along the generated trajectory, and the lateral distance to the target point. The positions of the control points are then optimized using the SQP algorithm to generate the optimal path.

A comparative analysis was conducted across four scenarios, validated using an integrated simulation environment of IPG CarMaker and MATLAB/Simulink. The comparison focused on the presence or absence of predicted information and the use of quartic and quintic Bézier algorithms. The results revealed that the combination of predicted information and the quintic Bézier algorithm produced the most stable and efficient paths, as indicated by metrics such as yaw rate, lateral acceleration, and steering wheel angle, which serve as indicators of lateral stability.

The proposed method leverages predicted information from surrounding vehicles to enhance collision avoidance and demonstrate that stable behavior can be maintained even in complex environments. However, there are situations where collision avoidance cannot be achieved through path planning alone. Therefore, future research should focus on integrating path and velocity planning based on predicted information, as the ego vehicle’s velocity also plays a critical role in ensuring safety.

## Figures and Tables

**Figure 1 sensors-24-07292-f001:**
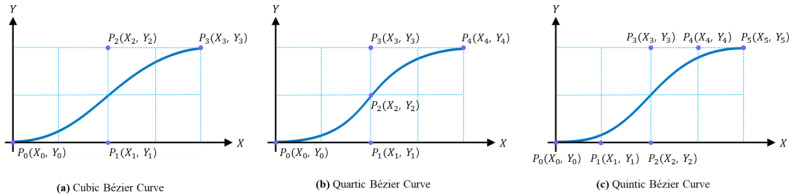
Example paths based on cubic, quartic, and quintic Bézier curves.

**Figure 2 sensors-24-07292-f002:**
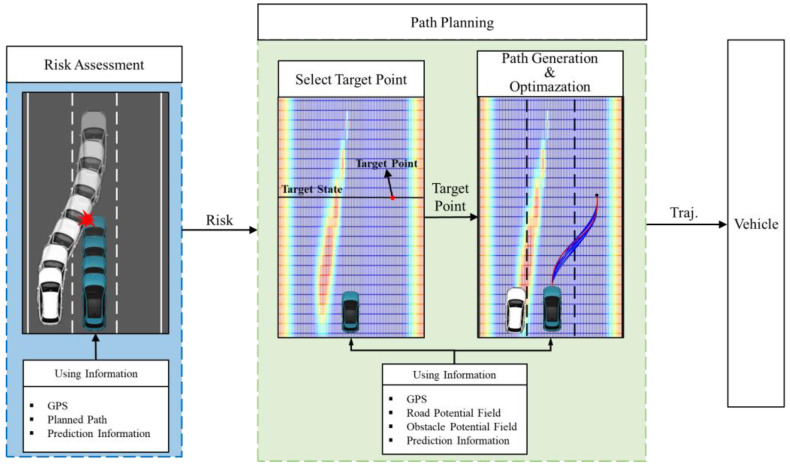
Architecture diagram of the collision-avoidance path planning system.

**Figure 3 sensors-24-07292-f003:**
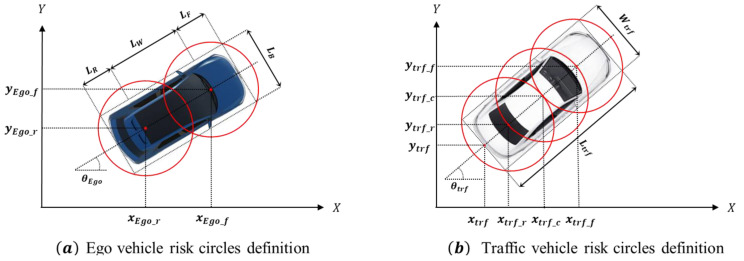
Schematic diagram of risk assessment circles.

**Figure 4 sensors-24-07292-f004:**
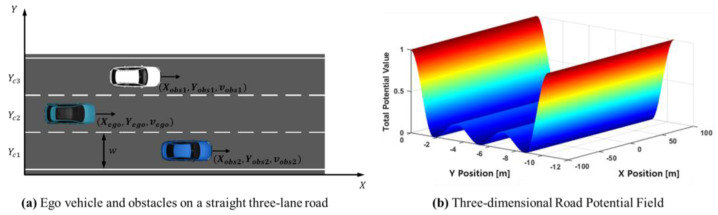
Illustration of the highway driving scenario and corresponding road potential field.

**Figure 5 sensors-24-07292-f005:**
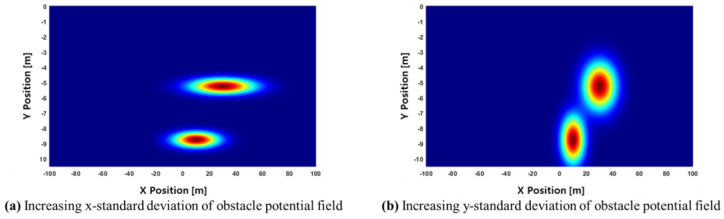
Comparison of obstacle potential field distributions under different conditions.

**Figure 6 sensors-24-07292-f006:**
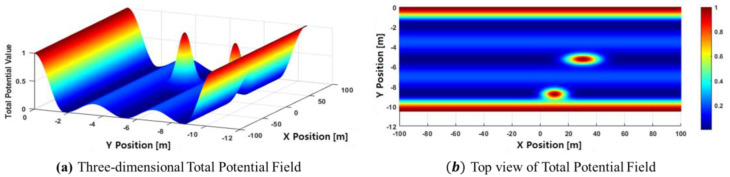
Visualization of the total potential field.

**Figure 7 sensors-24-07292-f007:**
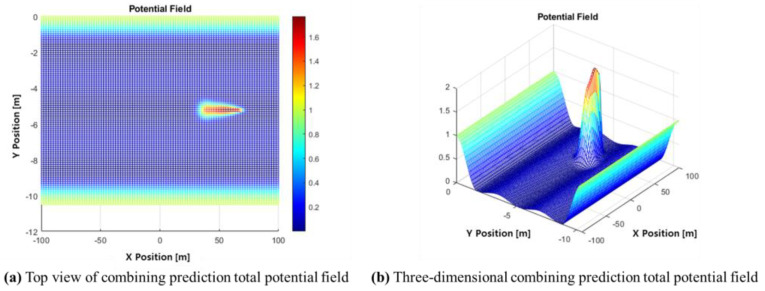
Visualization of the total potential field combining prediction.

**Figure 8 sensors-24-07292-f008:**
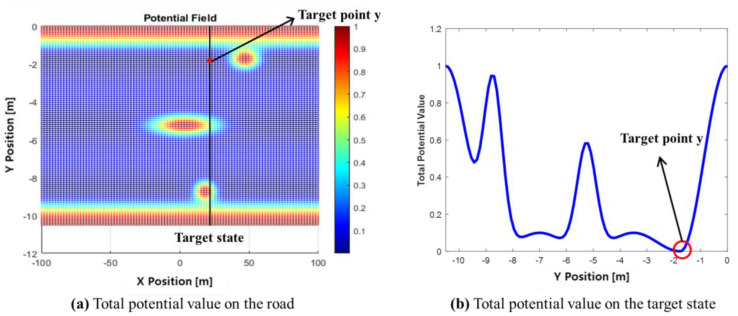
The total potential value of target positions.

**Figure 9 sensors-24-07292-f009:**
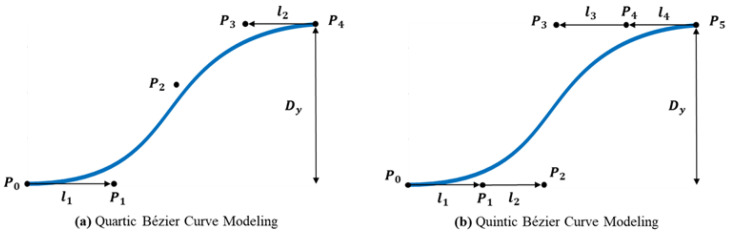
Quartic and quintic Bézier curve modeling.

**Figure 10 sensors-24-07292-f010:**
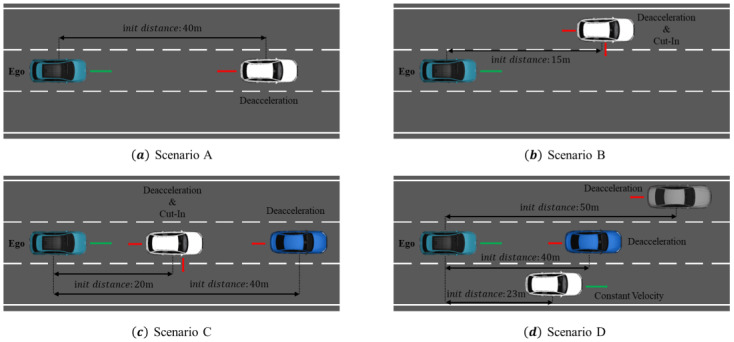
Scenarios used in the verification simulation.

**Figure 11 sensors-24-07292-f011:**
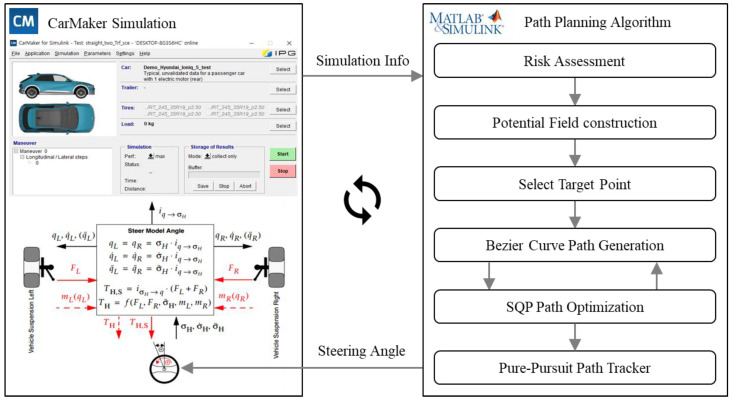
CarMaker Simulation and MATLAB/Simulink integrated environment for the collision-avoidance path planning methodology.

**Figure 12 sensors-24-07292-f012:**
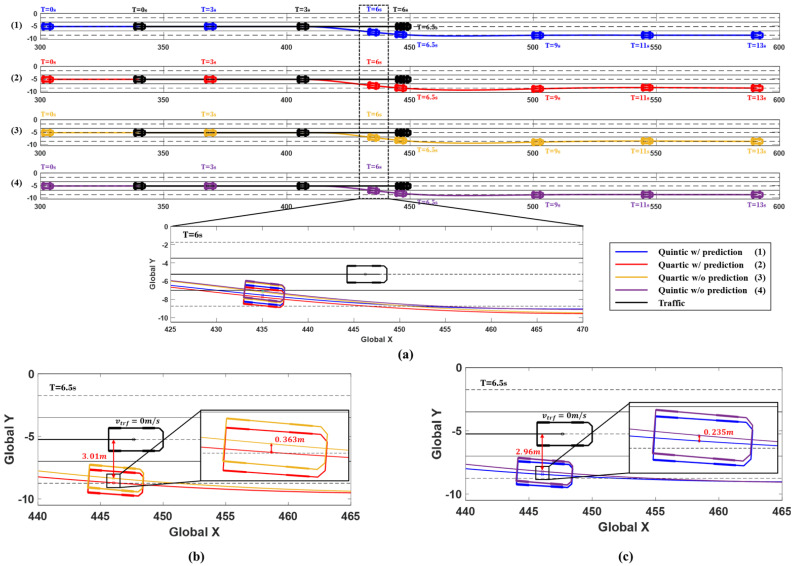
Simulation results for Scenario A. (**a**) Vehicle trajectories and time series vehicles position (0, 3, 6, 6.5, 9, 11, and 13 s). (**b**) Comparison of 6.5 s trajectories: Quartic w/prediction (red) vs. w/o prediction (yellow). (**c**) Comparison of 6.5 s trajectories: Quintic w/prediction (blue) vs. w/o prediction (purple).

**Figure 13 sensors-24-07292-f013:**
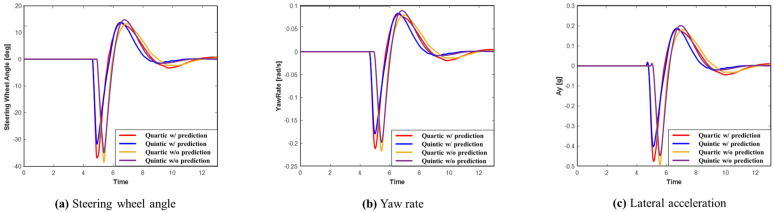
Scenario A: Comparison of steering wheel angle, yaw rate, and lateral acceleration.

**Figure 14 sensors-24-07292-f014:**
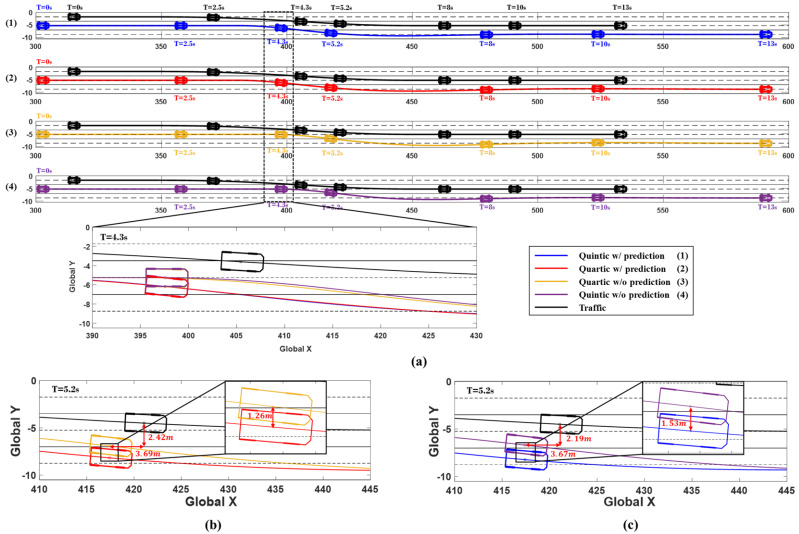
Simulation results for Scenario B. (**a**) Vehicle trajectories and time series vehicle positions (0, 2.5, 4.3, 5.2, 8, 10, and 13 s). (**b**) Comparison of 5.2 s trajectories: Quartic w/prediction (red) vs. w/o prediction (yellow). (**c**) Comparison of 5.2 s trajectories: Quintic w/prediction (blue) vs. w/o prediction (purple).

**Figure 15 sensors-24-07292-f015:**
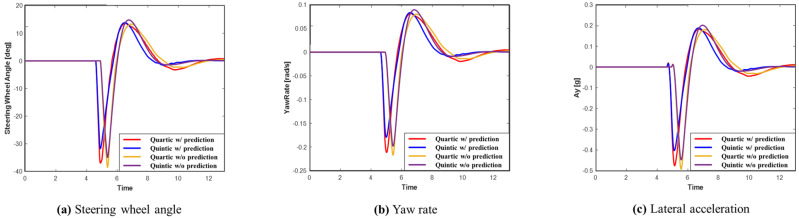
Scenario B: comparison of steering wheel angle, yaw rate, and lateral acceleration.

**Figure 16 sensors-24-07292-f016:**
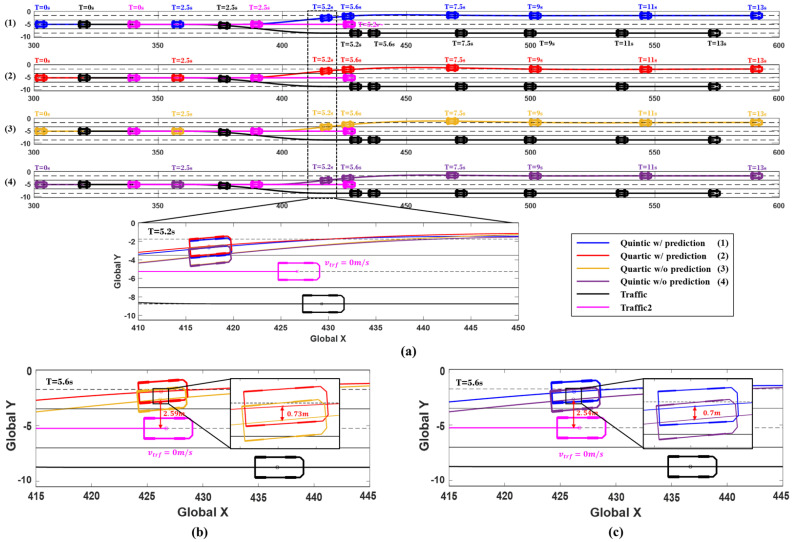
Simulation results for Scenario C. (**a**) Vehicle trajectories and time series vehicles position (0, 2.5, 5.2, 7.5, 9, 11, and 13 s). (**b**) Comparison of 5.6 s trajectories: Quartic /w prediction (red) vs. w/o prediction (yellow). (**c**) Comparison of 5.6 s trajectories: Quintic w/prediction (blue) vs. w/o prediction (purple).

**Figure 17 sensors-24-07292-f017:**
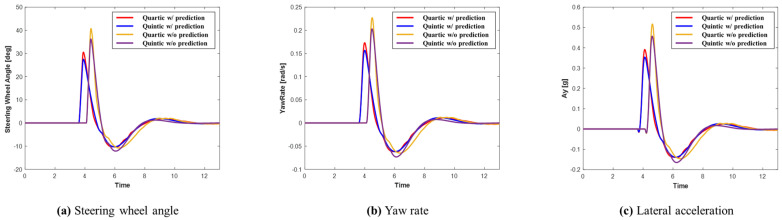
Scenario C: Comparison of steering wheel angle, yaw rate, and lateral acceleration.

**Figure 18 sensors-24-07292-f018:**
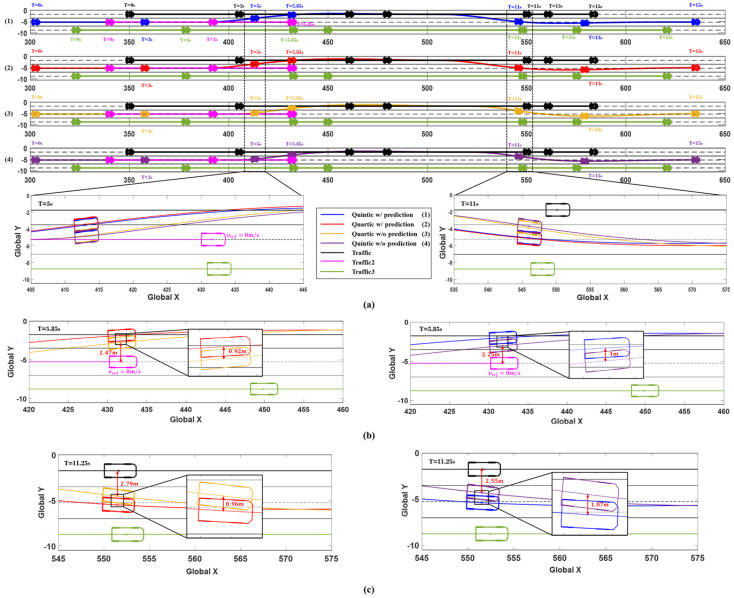
Simulation results of Scenario D. (**a**) Vehicle trajectories and time series vehicle position (0, 3, 5, 5.85, 11, 13, and 15 sec). (**b**) Comparison of 5.85 s trajectories: [Left] Quartic w/prediction (red) vs. w/o prediction (yellow), [Right] Quintic w/prediction (blue) vs. w/o prediction (purple). (**c**) Comparison of 11.25 s trajectories: [Left] Quartic w/prediction (red) vs. w/out prediction (yellow), [Right] Quintic w/prediction (blue) vs. w/o prediction (purple).

**Figure 19 sensors-24-07292-f019:**
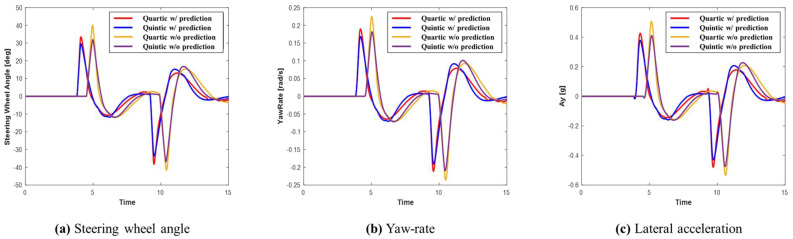
Scenario D: Comparison of steering wheel angle, yaw rate, and lateral acceleration.

**Table 1 sensors-24-07292-t001:** Parameters settings for risk circles.

$L_{F}$	$L_{w}$	$L_{R}$	$L_{B}$	$L_{trf}$	$W_{trf}$	Unit
0.845	2.97	0.82	1.89	4.47	1.97	m

**Table 2 sensors-24-07292-t002:** Parameters settings for the road and obstacle potential fields.

Symbol	Description	Value	Unit
w	Width of each lane	3.5	m
$Y_{c1}$	Global lateral position of the first lane	−1.75	m
$Y_{c3}$	Global lateral position of the third lane	−8.75	m
p	Parameter for the road potential field function	0.1	--
$\delta_{x}$	Longitudinal deviation for obstacle potential field	5	--
$\delta_{y}$	Lateral deviation for obstacle potential field	0.5	--
μ	Parameter to prevent errors in the obstacle potential field	1 × 10^−5^	--

**Table 3 sensors-24-07292-t003:** Hyperparameters settings for the objective function.

Symbol	Description	Value	Unit
a	Coefficient of curvature term for objective function	1	--
b	Coefficient of total potential value term for objective function	1	--
c	Coefficient of jerk term for objective function	1	--
$\kappa_{max}$	Maximum curvature	0.3	$m^{-1}$
γ	Threshold of TTC	2	s
ε	Coefficient of the laterally offset term for objective function	5	--

**Table 4 sensors-24-07292-t004:** Scenario A: Comparison of lateral stability based on maximum lateral acceleration, maximum yaw rate, and maximum steering wheel angle.

Scenario A	Maximum Lateral Acceleration (g)	Maximum Yaw Rate (deg/s)	Maximum Steering Wheel Angle (deg)
Quartic w/prediction	0.4758	0.2114	36.9478
**Quintic w/prediction**	**0.4030**	**0.1794**	**31.7464**
Quartic w/o prediction	0.5510	0.2449	44.6391
Quintic w/o prediction	0.5008	0.2222	39.9116

**Table 5 sensors-24-07292-t005:** Scenario B: Comparison of lateral stability based on maximum lateral acceleration, maximum yaw rate, and maximum steering wheel angle.

Scenario B	Maximum Lateral Acceleration (g)	Maximum Yaw Rate (deg/s)	Maximum Steering Wheel Angle (deg)
Quartic w/prediction	0.4716	0.2090	37.2937
**Quintic w/prediction**	**0.4213**	**0.1865**	**33.0276**
Quartic w/o prediction	0.5574	0.2429	44.6627
Quintic w/o prediction	0.5054	0.2247	40.1564

**Table 6 sensors-24-07292-t006:** Scenario C: Comparison of lateral stability based on maximum lateral acceleration, maximum yaw rate, and maximum steering wheel angle.

Scenario C	Maximum Lateral Acceleration (g)	Maximum Yaw Rate (deg/s)	Maximum Steering Wheel Angle (deg)
Quartic w/prediction	0.3914	0.1731	30.5570
**Quintic w/prediction**	**0.3539**	**0.1566**	**27.4941**
Quartic w/o prediction	0.5169	0.2271	40.7668
Quintic w/o prediction	0.4570	0.2030	36.2411

**Table 7 sensors-24-07292-t007:** Scenario D: Comparison of lateral stability based on maximum lateral acceleration, maximum yaw rate, and maximum steering wheel angle.

Scenario D	Maximum Lateral Acceleration (g)	Maximum Yaw Rate (deg/s)	Maximum Steering Wheel Angle (deg)
Quartic w/prediction	0.4802	0.2115	38.2716
**Quintic w/prediction**	**0.4300**	**0.1908**	**33.7257**
Quartic w/o prediction	0.5956	0.2601	46.2612
Quintic w/o prediction	0.5242	0.2327	41.6060

## Data Availability

Data are contained within the article.

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
