# Peer review of "Collision Avoidance Path Planning for Automated Vehicles Using Prediction Information and Artificial Potential Field"

_sensors, 2024, doi:10.3390/s24227292_

Round 1
Reviewer 1 Report
Comments and Suggestions for Authors
To address the issue of vehicles getting trapped in local minima during path planning, this study proposes integrating prediction data into the artificial potential field (APF) and optimizing the control points of the Quintic Bézier Curve using sequential quadratic programming. The effectiveness of this method is validated through simulations conducted in MATLAB. Overall, the paper is well-structured and the simulation experiments are comprehensive; however, there are still some areas for improvement.
Here are the corrections needed in the article:
1.All mathematical symbols and letters in the text should be written in a mathematical environment. For example, in line 94, n+1 should be changed to $n+1$.
2.In Figures 4, 5, and 6, the lengths and widths appear to be negative, which seems contradictory to our common sense. If these represent coordinates, please clarify.
3.In line 283, the reference to Figure 8.b appears to be incorrect.
4.In Equation 14, please clarify the meaning of the last bracketed mathematical symbol.
5.In Equation 16, TTC should not be in italics.
Here are some suggestions and questions regarding the article:
1.There are numerous parameters present in the equations throughout the article. Why did the authors choose the specific values given for these parameters? If there are reasons for these choices, please provide them.
2.In real-life situations, these parameters are often not fixed. How can these parameters be determined? Could the authors provide a framework for selecting these parameters?
3.Integrating predictive information into the APF method is likely to increase computational burden. Could this computational burden affect the accuracy of path prediction?
Reviewer 2 Report
Comments and Suggestions for Authors
1.Why is the trajectory prediction length set to 2 seconds? What is the basis for this choice?
2.The premise of this method is the accurate prediction of other vehicles' trajectories. What methods are used to predict the trajectories of other vehicles?
3.Does the trajectory prediction for other vehicles and the path planning for our vehicle take into account changes in speed?
Reviewer 3 Report
Comments and Suggestions for Authors
This paper proposes a path-planning method that leverages prediction information about surrounding objects to avoid collisions. The proposed method integrates the IPG CarMaker simulation environment and the MATLAB Simulink to perform real-time path planning, allowing the autonomous vehicle to avoid collisions efficiently. The approach combines artificial potential field (APF) with Bézier curve-based path generation and sequential quadratic programming (SQP) optimization to create an optimal, stable collision-avoidance path. Control inputs are seamlessly delivered from MATLAB Simulink to the CarMaker simulation, enabling real-time execution. Future research should focus on integrating path and velocity planning based on predicted information, as the ego vehicle's velocity also plays a critical role in ensuring safety. However, I considered it can be published in Sensors after a minor revision:
1. The introduction can further highlight the innovation of this paper. For example, the shortcomings of existing methods and the advantages of this research method are supplemented to highlight the necessity of research. In the literature review section, segmented simplification can be considered to further clearly show the existing methods and advantages and disadvantages of APF, Bezier curve, and vehicle trajectory prediction.
2. The images and tables in the article need further revision. The small font in the image can not let the reader see the information expressed in the image clearly, it is suggested that the author should modify it.
3. Explanations of the experimental environment and results can be added to the titles and notes of diagrams such as architecture diagrams and path generation diagrams to make them more intuitive. add some references such as“J. Mater. Res. Technol., 2024, 29: 5667-5680., Materials Today Energy, 2024, 44: 101626., Journal of Colloid and Interface Science, 2023, 630: 86-98. , Chemical Engineering Journal, 2024, 492: 152245., Chemical Engineering Journal, 2024, 491: 151862., Chemical Engineering Journal, 2024, 491: 152041., Chem. Eng. J. 2024, 500: 157119.”
4. The author uses a large number of formulas to illustrate the content of this paper. Among them, the author needs to pay attention to the format of the formulas to ensure that the fonts of all formulas are consistent and aligned up and down, and all formulas look complete. In addition, the author should give the concept of some important physical quantities in the formula, which is easier for readers to understand.
5. There are few references in this paper, so the author can cite more new and influential articles to support the content of this paper. In addition, the format of references should be paid attention to ensure the consistency of the format.
Comments on the Quality of English Languageminor
